# Attention Sinks as Internal Signals
# for Hallucination Detection in Large Language Models

**Jakub Binkowski** [1]  **Kamil Adamczewski** [1]  **Tomasz Kajdanowicz** [1]

## Abstract

Large language models frequently exhibit hallucinations: fluent and confident outputs that are factually incorrect or unsupported by the input context. While recent hallucination detection methods have explored various features derived from attention maps, the underlying mechanisms they exploit remain poorly understood. In this work, we propose SinkProbe, a hallucination detection method grounded in the observation that hallucinations are deeply entangled with attention sinks—tokens that accumulate disproportionate attention mass during generation—indicating a transition from distributed, input-grounded attention to compressed, prior-dominated computation. Importantly, although sink scores are computed solely from attention maps, we find that the classifier preferentially relies on sinks whose associated value vectors have large norms. Moreover, we show that previous methods implicitly depend on attention sinks by establishing their mathematical relationship to sink scores. Our findings yield a novel hallucination detection method grounded in theory that produces state-of-the-art results across popular datasets and LLMs.

## 1. Introduction

Large language models (LLMs) have achieved remarkable success across a wide range of natural language understanding and generation tasks, and are frequently deployed in settings that require factual reliability, such as question answering, summarization, and decision support (Kwiatkowski et al., 2019; Lin et al., 2022; Zhao et al., 2025). Despite these advances, LLMs remain prone to *hallucinations*—fluent and confident outputs that are factually incorrect, unverifiable,

or unsupported by the input context (Orgad et al., 2025; Farquhar et al., 2024). Hallucinations pose a fundamental challenge to the safe and trustworthy use of LLMs. Unlike traditional task-specific models, modern LLMs operate in open-ended regimes, where errors often manifest as plausible fabrications rather than explicit contradictions (Huang et al., 2025).

A number of existing approaches address hallucinations at the *output level*. These include fact-checking against external knowledge sources, retrieval-augmented generation, consistency checks across multiple sampled responses, and uncertainty-based metrics derived from token probabilities or entropy (Ren et al., 2023; Manakul et al., 2023; Farquhar et al., 2024; Fadeeva et al., 2024; Sawczyn et al., 2026). While effective in certain settings, such methods typically require additional resources such as external corpora, or multiple generations, and provide limited insight into the internal computational mechanisms that give rise to hallucinations.

Recently, a complementary line of work has explored *internal signals* of hallucination by analyzing attention maps, hidden representations, or spectral properties of Transformer Decoder models (Vaswani et al., 2017). These methods exploit the observation that hallucinated generations are often associated with atypical internal representations (Chen et al., 2024; Du et al., 2024), lack of grounding of generations (Chuang et al., 2024; Bazarova et al., 2025), or abnormal attention dynamics (Sriramanan et al., 2024; Binkowski et al., 2025; Frasca et al., 2026). In this work, we trace hallucinations to breakdowns in internal information flow captured by attention sink scores—which measure the disproportionate allocation of attention to initial placeholder tokens used for model stabilization. We show that this perspective both unifies prior detection methods and yields a simple and effective hallucination detector.

*Attention sinks* (Xiao et al., 2024; Gu et al., 2025), tokens attracting disproportionate attention despite low semantic relevance, are a pervasive mechanism for compression and information routing in Transformers (Barbero et al., 2025; Ruscio et al., 2025). We hypothesize that hallucinations may stem from breakdowns in this internal flow, rather than knowledge gaps alone. Consequently, we leverage

---

[1]Department of Artificial Intelligence, Wroclaw University of Science and Technology. Correspondence to: Jakub Binkowski <jakub.binkowski@pwr.edu.pl>.

*Proceedings of the 43rd International Conference on Machine Learning*, Seoul, South Korea. PMLR 306, 2026.

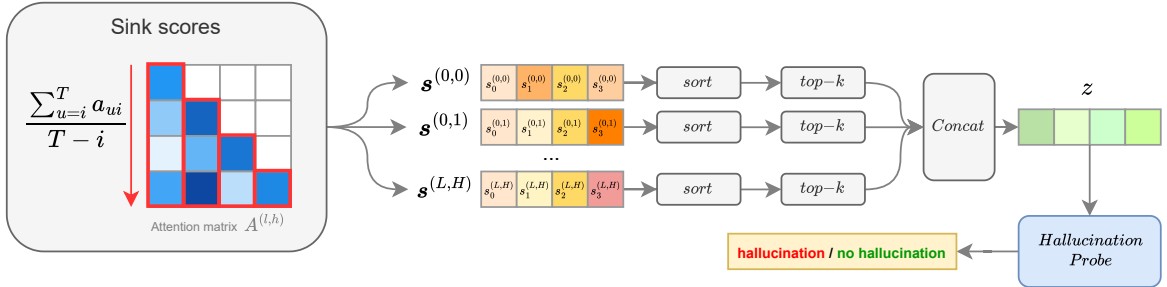

*Figure 1.* Pipeline for hallucination detection based on attention sink scores. For each layer $l$ and head $h$, we compute sink scores $\mathbf{s}^{(l,h)}$, defined as the average of attention scores directed toward each token position. These scores are then sorted, and the top-$k$ values are selected as features. The selected features from all layers and heads are concatenated to form the final feature vector $\mathbf{z}$, which is passed to a hallucination probe – a binary classifier trained on labeled examples to distinguish hallucinated outputs from factually correct ones at inference time.

the *attention sink score*: an interpretable diagnostic derived from cross-layer attention statistics. This yields a compact, token-agnostic signal for detecting hallucinations.

Importantly, we show that not all tokens with large attention are equally relevant for hallucination detection. While sink scores are computed solely from attention maps, their computational impact depends on the associated value vectors. We find that the hallucination signal is concentrated in tokens where high attention concentration coincides with unusually large value norms—a subset of computationally active tokens that dominate the attention output and induce compressed, prior-dominated representations.

We further provide a unifying perspective on existing attention-based hallucination detectors, demonstrating that several spectral and graph-based methods can be interpreted as transformations of sink behavior. Across a wide range of models and benchmarks, sink-score–based features consistently yield superior results, outperforming other detectors within the same family in the vast majority of settings

In summary, our main contributions are as follows:

- We propose SinkProbe—a novel method for hallucination detection from attention maps.
- We identify computationally active sinks, showing that sink-based signals are strongest when attention concentration coincides with large value vector norms.
- We unify existing attention-based detectors by relating them to sink scores.
- We show that sink-score-based features achieve state-of-the-art hallucination detection across several common LLMs and benchmarks in vast majority of settings.

The implementation is publicly available at github.com/graphml-lab-pwr/sink-probe.

## 2. Method

We leverage the attention sink score, an attention-based measure derived from attention maps that quantifies the concentration of attention on the tokens. Our method operates purely on internal attention weights and produces a compact feature representation suitable for hallucination detection. We call this method SinkProbe and introduce it in the following section by first formalizing sink scores at the token, head, and layer levels, then describing how they are aggregated into a probe feature vector. The method is summarized in Figure 1.

### 2.1. Attention Sinks

To trace the information flow within LLMs, we leverage *attention sink score*—a measure primarily developed to detect attention sinks (Xiao et al., 2024; Gu et al., 2025), i.e., tokens that accumulate significant attention from future contexts while contributing minimal semantic value. Let $\mathbf{A}^{(l,h)} \in \mathbb{R}^{T \times T}$ denote the attention matrix produced by attention head $h \in \{1, \ldots, H\}$ in layer $l \in \{1, \ldots, L\}$, where $\mathbf{A}^{(l,h)}_{u,i}$ represents the attention weight from token $u$ to token $i$, with $u, i \in \{1, \ldots, T\}$, and $T$ is the sequence length. The matrix $\mathbf{A}^{(l,h)}$ is row-stochastic and strictly causal, i.e., $\mathbf{A}^{(l,h)}_{u,i} = 0$ for all $i > u$.

**Definition 2.1** (Sink Score, Gu et al., 2025). For a given attention head $(l, h)$, the sink score of token $i$ is defined as

$$s_i^{(l,h)} = \frac{1}{T-i} \sum_{u=i}^{T} \mathbf{A}_{u,i}^{(l,h)}. \tag{1}$$

This quantity measures the average amount of attention token $i$ receives from all subsequent tokens, normalized by the remaining sequence length. High sink scores indicate tokens that persistently attract attention as generation progresses. Sink scores are defined independently for each layer and head, yielding a tensor $S \in \mathbb{R}^{L \times H \times T}$.

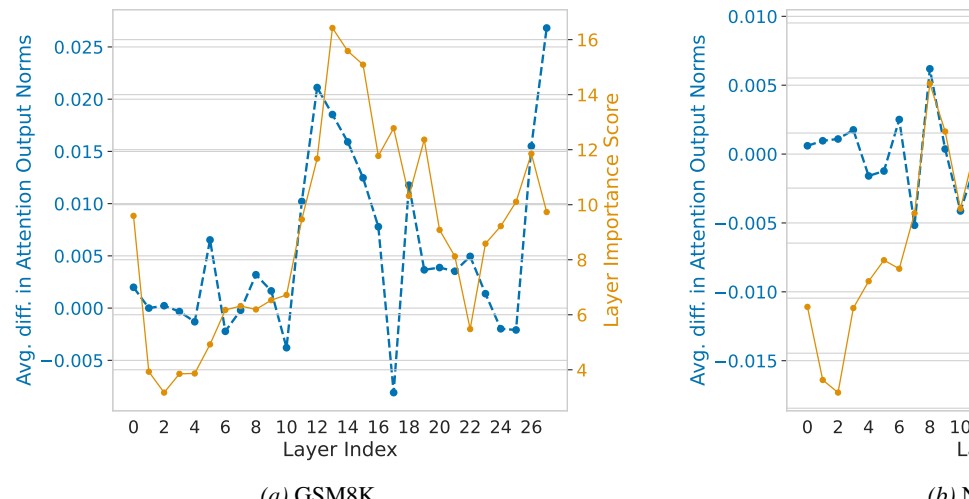

*(a)* GSM8K            *(b)* NQ-Open

*Figure 2.* Relationship between attention output norm differences and layer importance scores for Llama3.2-3B-Instruct on GSM8K (a) and NQ-Open (b). The blue dashed line shows the average difference in attention output norms between hallucinated and non-hallucinated examples, i.e., $\text{Avg}\left(\|O_u^{(l)}(\text{hallu})\|_2 - \|O_u^{(l)}(\text{non-hallu})\|_2\right)$, with error bars indicating standard error. The orange solid line shows layer importance scores from the hallucination probe (obtained from regularized logistic regression model, see Section 4.1 for details). Both datasets exhibit alignment in middle layers, where norm differences and importance scores peak together.

### 2.2. From Sink Scores to Features

Sink scores typically exhibit a highly skewed distribution: in most attention heads, only a small number of tokens accumulate large scores, while the majority receive negligible attention. Rather than relying on token identity or position, we summarize this structure using order statistics.

For each head $(l, h)$, we sort the sink scores $\tilde{s}^{(l,h)} = \text{sort}\left(s^{(l,h)}\right)$, and retain the top-$k$ values. The final feature vector for a given example is constructed by concatenating these values across all layers and heads:

$$z = \bigoplus_{l=1}^{L} \bigoplus_{h=1}^{H} \left[\tilde{s}_T^{(l,h)}, \tilde{s}_{T-1}^{(l,h)}, \dots, \tilde{s}_{T-k+1}^{(l,h)}\right] \in \mathbb{R}^{L \cdot H \cdot k}. \quad (2)$$

This representation captures the degree of attention concentration while remaining agnostic to token semantics and sequence length. To evaluate the predictive power of sink scores, we train a lightweight hallucination probe, i.e., logistic regression classifier.

### 2.3. Role of Value Vectors in Attention Sink Behavior

While attention sink scores characterize the structural concentration of attention weights in Transformer Decoder, their effect on the model's computation depends critically on the associated value vectors. For a given attention head $(l, h)$, the output at token position $u$ is given by

$$\mathbf{O}_u^{(l,h)} = \sum_{i=1}^{u} \mathbf{A}_{u,i}^{(l,h)} \mathbf{V}_i^{(l,h)}$$

A token $i$ acting as an attention sink contributes to this output in proportion to both its incoming attention mass and the magnitude of its value vector. Attention sinks, which serve to prevent over-mixing (Barbero et al., 2025) and are often associated with semantically vacuous tokens (e.g., the $\langle \text{bos} \rangle$ token), have been shown to exhibit small value-vector magnitudes (Gu et al., 2025). Consequently, a sink with small $\|\mathbf{V}_i^{(l,h)}\|$ exerts limited influence on $\mathbf{O}_u^{(l,h)}$, even when it attracts substantial attention, whereas a sink with a large value norm can dominate the attention output across multiple positions. This distinction implies that only a subset of attention sinks are *computationally active*—that is, they significantly influence hidden representations through repeated injection of high-magnitude value vectors.

Motivated by this observation, we examined the norms of attention outputs $\|\mathbf{O}_{u,i}^{(l,h)}\|_2$. Specifically, we traced these output vectors and computed their norms for tokens identified as important by our hallucination probe (see Section 4.1), partitioning the data according to whether each example was hallucinated or non-hallucinated. Figure 2 presents the aggregated norm differences alongside probe importance scores for two selected datasets, stratified by layer. The observed pattern suggests that sink-score–based features primarily capture sinks whose attention concentration coincides with elevated magnitudes, rendering them more likely to exert substantial influence on the attention output. Although this association is not universal—as evidenced by divergent behavior in certain layers—it provides a principled account of why only a subset of attention sinks consistently contributes to hallucination detection, despite the

widespread presence of sink behavior across heads and layers. We note that this observation does not imply direct causality. Future interventional studies will be essential to verify the underlying mechanisms of this phenomenon.

## 2.4. Sink Score as an Underlying Concept for Hallucination Detection Methods

In this section, we show that several existing attention-based hallucination detection methods can be interpreted through the lens of attention sink behavior. Although the concepts underlying these methods were originally developed from different motivations and formulated using distinct mathematical constructions, we demonstrate that they implicitly rely on the concentration of attention onto sink tokens. This perspective provides a unifying explanation for their effectiveness and clarifies the role of attention collapse as a common underlying signal.

**LLMCheck (Sriramanan et al., 2024)** was one of the first methods to leverage attention maps for hallucination detection, proposing AttentionScore, which aggregates per-layer self-attention terms into a scalar hallucination indicator. Specifically, for a given layer $l$, the score is defined as an average log-determinant of its attention maps:

$$\text{AttentionScore}^{(l)} = \frac{1}{HT} \sum_{h=1}^{H} \sum_{i=1}^{T} \log(\mathbf{A}_{ii}^{(l,h)}) \quad (3)$$

If a token has large sink scores, i.e., later tokens attend to it heavily, then their attention mass is concentrated outside the diagonal $\mathbf{A}_{ii}^{(l,h)}$, thus decreasing the log-determinant $\frac{1}{T} \sum_{i=1}^{T} \log(\mathbf{A}_{ii}^{(l,h)})$. Therefore, the tokens absorbing a substantial fraction of the attention mass disproportionately affect this score relative to non-sink tokens.

**LookbackLens (Chuang et al., 2024)** proposed to detect contextual hallucinations by comparing attention allocated to the prompt versus attention allocated to the already generated response. The key assumption is that hallucinated generations are less grounded in the prompt, particularly in retrieval-augmented generation (RAG) settings. The central quantity is the lookback ratio. Let $|P|$ be the number of prompt tokens and $|R|$ be the number of response tokens. For each generated token $i$, define:

$$\text{Attn}_{\text{ctx}}^{(l,h)}(i) = \frac{1}{|P|} \sum_{u=1}^{|P|} \mathbf{A}_{iu}^{(l,h)}$$

$$\text{Attn}_{\text{resp}}^{(l,h)}(i) = \frac{1}{|R|} \sum_{u=|P|+1}^{|P|+|R|} \mathbf{A}_{iu}^{(l,h)}$$

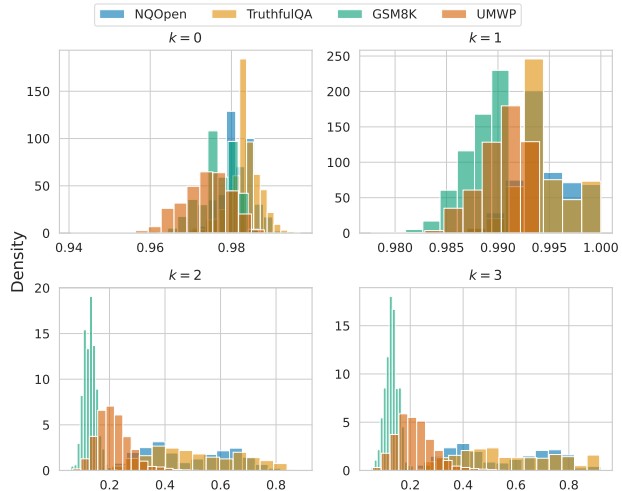

*Figure 3.* Average frequency that the $k$-th token–ranked by sink score–appears in the prompt, averaged across all attention heads for Llama 3.2-3B.

Then, the lookback ratio $LR_i^{(l,h)}$ for a layer $l$ and head $h$ is defined as:

$$LR_i^{(l,h)} = \frac{\text{Attn}_{\text{resp}}^{(l,h)}(i)}{\text{Attn}_{\text{ctx}}^{(l,h)}(i) + \text{Attn}_{\text{resp}}^{(l,h)}(i)} \quad (4)$$

Attention sink scores capture complementary, column-wise structure by identifying tokens that consistently attract large amounts of attention across decoding steps. Although sink scores and lookback ratios are not algebraically related, they are coupled through the row-stochastic normalization of attention. When attention concentrates on a small set of sink tokens at a given step, the remaining tokens necessarily receive less attention.

This coupling has different implications depending on the location of the sinks. Sinks among generated tokens promote self-referential attention, increasing $\text{Attn}_{\text{resp}}^{(l,h)}$ and the lookback ratio, and are therefore closely associated with hallucinated outputs. In contrast, sinks located in the prompt concentrate attention on a narrow subset of prompt tokens without ensuring effective use of the full context. Consequently, prompt sinks reflect attention collapse rather than reliable grounding and do not consistently correspond to reduced hallucination risk. This distinction clarifies why lookback-based detectors are primarily sensitive to generated-token attention collapse, while sink-based diagnostics capture a broader class of attention concentration phenomena.

Subsequently, we analyze which parts of the input (prompt vs. response) fall within our proposed top-$k$ sink scores. Figure 3 shows that the highest-ranked sink ($k$=0) lies almost exclusively in the prompt across all datasets (mass near 1), consistent with it typically corresponding to ⟨bos⟩. The next sink ($k$=1) also falls predominantly in the prompt, with

modest dataset-dependent variability. In contrast, lower-ranked sinks ($k \geq 2$) are increasingly likely to occur in the generated answer; this shift is particularly pronounced on GSM8K (and to a lesser extent UMWP), whereas NQ-Open and TruthfulQA retain a substantial prompt contribution. This finding further highlights that later sinks, which are crucial for hallucination detection, often originate from the generated answer rather than purely from the prompt (see Section 4.2 for a study of the effect of $k$). Extended semantic analysis of the tokens is presented in Section D.

**Topological Divergence (Bazarova et al., 2025)**  introduced the TOHA method, which, similarly to Lookback-Lens, relies on the assumption that there is a relationship between prompt and response tokens (the response should be grounded in the prompt) and was designed specifically for RAG scenarios.

TOHA formulates the attention map as a weighted graph with edge weights representing a pseudo-distance between tokens, e.g., $d_{ij} = 1 - a_{ij}$, where $a_{ij}$ denotes the attention weight between token $j$ and token $i$. Similarly to Look-backLens, tokens are partitioned into prompt tokens $P$ and response tokens $R$. The method computes probe features based on topological divergence, which in this setting can be reduced to computing the minimum spanning tree (MST) between a collapsed prompt node and the response tokens:

$$\text{MTopDiv}(P, R) = \sum_{(i,j) \in \text{MST}(P,R)} d_{ij} \tag{5}$$

Since sink nodes can induce low pseudo-distances, they are likely to appear in the MST and thus can play a significant role in detection, as also observed in the original work.

**Spectral features (Binkowski et al., 2025)**  proposed leveraging attention graphs and order statistics of graph Laplacian eigenvalues to detect hallucinations, motivated by the observation that the Laplacian can capture disruptions to information flow within the LLM. Surprisingly, we find that these eigenvalues correspond to sink scores discounted by self-attention. Specifically, given the lower-triangular structure of the attention and Laplacian matrices, the eigenvalues are simply the entries on the main diagonal:

$$l_{ii}^{(l,h)} = d_{ii}^{(l,h)} - a_{ii}^{(l,h)} = \frac{\sum_{u=1}^{T} a_{ui}^{(l,h)}}{T - i} - a_{ii}^{(l,h)}$$

By noticing the direct correspondence of the first term to the sink score, we can rewrite the eigenvalues as:

$$l_{ii}^{(l,h)} = s_i^{(l,h)} - a_{ii}^{(l,h)}$$

This relationship reveals that Laplacian-based features inherently encode sink score information while additionally discounting self-attention.

# 3. Experiments

## 3.1. Experimental Setup

**Datasets.**  We evaluate our method on seven diverse hallucination detection benchmarks spanning question answering and mathematical reasoning: *GSM8K* (Cobbe et al., 2021) contains grade school math word problems requiring multi-step reasoning; *UMWP* (Sun et al., 2024) provides university-level math word problems designed to probe model knowledge boundaries; *TruthfulQA* (Lin et al., 2022) tests factual accuracy on questions where models commonly produce misconceptions; *TriviaQA* (Joshi et al., 2017) and *NQ-Open* (Kwiatkowski et al., 2019) evaluate open-domain factual knowledge; *SQuADv2* (Rajpurkar et al., 2018) includes reading comprehension questions; and *HaluEvalQA* (Li et al., 2023) provides a targeted hallucination evaluation benchmark. This varied selection ensures our evaluation covers closed-book factual QA, passage-grounded reading comprehension, and mathematical reasoning. Further details on the datasets and their preprocessing are provided in Section B.1.

**LLMs.**  We evaluate our method on four widely used open-weight LLMs spanning three model families and ranging from 3B to 12B parameters: Llama3.2-3B (AI@Meta, 2024), Phi3.5 (4B) (Abdin et al., 2024), Llama3.1-8B (AI@Meta, 2024), and Mistral-Nemo (12B) (Mistral AI Team & NVIDIA, 2024). Further details about the models are provided in Section B.2.

**Methodology.**  We perform inference on each dataset, recording generated answers and attention maps. Model responses are evaluated against gold answers using an LLM-as-judge setting, with selective manual review to ensure label quality. Then, we extract features for SinkProbe and baselines, and train a logistic regression hallucination probe, reporting ROC-AUC metric from 5-fold cross-validation. Methodology and implementation details are provided in Section A. The implementation is available at github.com/graphml-lab-pwr/sink-probe.

**Baselines.**  We compare SinkProbe against several attention-based hallucination detection baselines: (1) AttentionScore (Sriramanan et al., 2024), an unsupervised method that computes the log-determinant of attention matrices aggregated across layers and heads, with ROC-AUC measured directly on raw scores; (2) AttnLogDet, a supervised counterpart of AttentionScore that uses attention log-determinants as features for a trained hallucination probe; (3) AttnEigvals and (4) LapEigval (Binkowski et al., 2025), which extract the top-$k$ eigenvalues of attention and Laplacian matrices, respectively; (5) LookbackLens (Chuang et al., 2024), which measures the ratio of attention attributed to context; and (6) MTopDiv (Bazarova et al., 2025), which

*Table 1.* Results of hallucination detection experiments, the values represent mean and standard deviation of ROC-AUC scores over 5-fold cross-validation, $\Delta$ represents relative improvement of SinkProbe compared with second best or best performing method.

| LLM | Dataset
Feature | GSM8K | HaluEvalQA | NQ-Open | SQuADv2 | TriviaQA | TruthfulQA | UMWP |
|---|---|---|---|---|---|---|---|---|
| Llama3.2-3B | AttnScore | $0.743 \pm 0.049$ | $0.737 \pm 0.015$ | $0.661 \pm 0.027$ | $0.637 \pm 0.027$ | $0.672 \pm 0.013$ | $0.667 \pm 0.041$ | $0.702 \pm 0.017$ |
| | AttnLogDet | $0.819 \pm 0.010$ | $0.817 \pm 0.012$ | $0.723 \pm 0.014$ | $0.739 \pm 0.009$ | $0.789 \pm 0.009$ | $\underline{0.771 \pm 0.048}$ | $\mathbf{0.809 \pm 0.017}$ |
| | AttnEigval | $0.802 \pm 0.025$ | $0.817 \pm 0.014$ | $0.732 \pm 0.024$ | $0.738 \pm 0.027$ | $0.798 \pm 0.017$ | $0.765 \pm 0.030$ | $0.704 \pm 0.025$ |
| | LapEigval | $0.825 \pm 0.009$ | $0.837 \pm 0.011$ | $0.736 \pm 0.012$ | $0.753 \pm 0.014$ | $0.824 \pm 0.017$ | $0.748 \pm 0.047$ | $0.720 \pm 0.010$ |
| | LookbackLens | $\underline{0.835 \pm 0.017}$ | $0.836 \pm 0.009$ | $0.743 \pm 0.020$ | $\underline{0.765 \pm 0.008}$ | $0.811 \pm 0.015$ | $0.741 \pm 0.072$ | $0.749 \pm 0.020$ |
| | MTopDiv | $0.816 \pm 0.026$ | $\underline{0.846 \pm 0.009}$ | $\underline{0.754 \pm 0.012}$ | $0.764 \pm 0.010$ | $\underline{0.827 \pm 0.018}$ | $0.757 \pm 0.044$ | $0.735 \pm 0.017$ |
| | SinkProbe | $\mathbf{0.845 \pm 0.017}$ | $\mathbf{0.847 \pm 0.008}$ | $\mathbf{0.758 \pm 0.016}$ | $\mathbf{0.769 \pm 0.010}$ | $\mathbf{0.835 \pm 0.017}$ | $\mathbf{0.785 \pm 0.059}$ | $\underline{0.757 \pm 0.017}$ |
| | $\Delta$ | +1.2% | +0.1% | +0.5% | +0.5% | +1.0% | +1.8% | -6.4% |
| Phi3.5 | AttnScore | $0.741 \pm 0.069$ | $0.712 \pm 0.009$ | $0.729 \pm 0.022$ | $0.678 \pm 0.015$ | $0.705 \pm 0.016$ | $0.693 \pm 0.045$ | $0.757 \pm 0.022$ |
| | AttnLogDet | $0.782 \pm 0.058$ | $0.805 \pm 0.006$ | $0.801 \pm 0.011$ | $0.760 \pm 0.016$ | $0.837 \pm 0.004$ | $\underline{0.774 \pm 0.044}$ | $0.864 \pm 0.023$ |
| | AttnEigval | $0.794 \pm 0.055$ | $0.803 \pm 0.004$ | $0.779 \pm 0.006$ | $0.750 \pm 0.023$ | $0.833 \pm 0.006$ | $0.770 \pm 0.025$ | $0.864 \pm 0.017$ |
| | LapEigval | $0.775 \pm 0.047$ | $0.822 \pm 0.006$ | $\underline{0.820 \pm 0.020}$ | $\underline{0.767 \pm 0.017}$ | $0.859 \pm 0.003$ | $0.755 \pm 0.046$ | $0.862 \pm 0.025$ |
| | LookbackLens | $0.843 \pm 0.028$ | $0.828 \pm 0.004$ | $0.816 \pm 0.015$ | $\underline{0.767 \pm 0.020}$ | $0.850 \pm 0.006$ | $0.773 \pm 0.043$ | $\underline{0.877 \pm 0.021}$ |
| | MTopDiv | $\underline{0.845 \pm 0.035}$ | $0.835 \pm 0.005$ | $0.818 \pm 0.022$ | $\mathbf{0.780 \pm 0.022}$ | $\underline{0.866 \pm 0.006}$ | $\mathbf{0.797 \pm 0.043}$ | $0.872 \pm 0.019$ |
| | SinkProbe | $\mathbf{0.854 \pm 0.021}$ | $\mathbf{0.846 \pm 0.004}$ | $\mathbf{0.821 \pm 0.011}$ | $0.764 \pm 0.019$ | $\mathbf{0.877 \pm 0.006}$ | $0.761 \pm 0.046$ | $\mathbf{0.896 \pm 0.015}$ |
| | $\Delta$ | +1.1% | +1.3% | +0.1% | -2.1% | +1.3% | -4.5% | +2.2% |
| Llama3.1-8B | AttnScore | $0.752 \pm 0.048$ | $0.770 \pm 0.012$ | $0.677 \pm 0.017$ | $0.656 \pm 0.019$ | $0.678 \pm 0.011$ | $0.704 \pm 0.026$ | $0.722 \pm 0.026$ |
| | AttnLogDet | $0.812 \pm 0.034$ | $0.849 \pm 0.015$ | $0.759 \pm 0.018$ | $0.752 \pm 0.022$ | $0.827 \pm 0.014$ | $0.765 \pm 0.041$ | $0.825 \pm 0.024$ |
| | AttnEigval | $0.797 \pm 0.037$ | $0.850 \pm 0.011$ | $0.759 \pm 0.029$ | $0.761 \pm 0.012$ | $0.835 \pm 0.008$ | $0.773 \pm 0.042$ | $0.818 \pm 0.021$ |
| | LapEigval | $\mathbf{0.826 \pm 0.029}$ | $0.878 \pm 0.009$ | $\underline{0.787 \pm 0.027}$ | $0.785 \pm 0.024$ | $\underline{0.874 \pm 0.013}$ | $0.757 \pm 0.068$ | $0.834 \pm 0.016$ |
| | LookbackLens | $0.816 \pm 0.042$ | $0.879 \pm 0.007$ | $0.776 \pm 0.024$ | $0.776 \pm 0.019$ | $0.868 \pm 0.012$ | $0.767 \pm 0.040$ | $\underline{0.838 \pm 0.017}$ |
| | MTopDiv | $0.803 \pm 0.036$ | $\underline{0.881 \pm 0.008}$ | $0.772 \pm 0.021$ | $\underline{0.787 \pm 0.018}$ | $0.874 \pm 0.009$ | $\underline{0.775 \pm 0.047}$ | $0.809 \pm 0.014$ |
| | SinkProbe | $\underline{0.824 \pm 0.025}$ | $\mathbf{0.890 \pm 0.005}$ | $\mathbf{0.789 \pm 0.026}$ | $\mathbf{0.798 \pm 0.020}$ | $\mathbf{0.883 \pm 0.012}$ | $\mathbf{0.778 \pm 0.042}$ | $\mathbf{0.879 \pm 0.009}$ |
| | $\Delta$ | -0.2% | +1.0% | +0.3% | +1.4% | +1.0% | +0.4% | +4.9% |
| Mistral-Nemo | AttnScore | $0.712 \pm 0.058$ | $0.688 \pm 0.013$ | $0.666 \pm 0.016$ | $0.641 \pm 0.019$ | $0.659 \pm 0.018$ | $0.678 \pm 0.053$ | $0.773 \pm 0.026$ |
| | AttnLogDet | $0.797 \pm 0.063$ | $0.797 \pm 0.006$ | $0.750 \pm 0.019$ | $0.749 \pm 0.013$ | $0.810 \pm 0.013$ | $0.798 \pm 0.048$ | $0.847 \pm 0.018$ |
| | AttnEigval | $0.780 \pm 0.062$ | $0.791 \pm 0.008$ | $0.730 \pm 0.026$ | $0.745 \pm 0.015$ | $0.815 \pm 0.014$ | $0.779 \pm 0.056$ | $0.826 \pm 0.017$ |
| | LapEigval | $0.804 \pm 0.049$ | $0.831 \pm 0.005$ | $0.765 \pm 0.028$ | $0.777 \pm 0.019$ | $0.860 \pm 0.003$ | $0.806 \pm 0.042$ | $0.847 \pm 0.007$ |
| | LookbackLens | $\mathbf{0.841 \pm 0.063}$ | $\underline{0.837 \pm 0.009}$ | $\underline{0.782 \pm 0.022}$ | $\underline{0.783 \pm 0.014}$ | $\underline{0.866 \pm 0.009}$ | $0.804 \pm 0.039$ | $\underline{0.866 \pm 0.017}$ |
| | MTopDiv | $\underline{0.813 \pm 0.062}$ | $\underline{0.837 \pm 0.009}$ | $0.776 \pm 0.025$ | $0.770 \pm 0.012$ | $0.865 \pm 0.007$ | $\underline{0.811 \pm 0.030}$ | $0.841 \pm 0.005$ |
| | SinkProbe | $0.811 \pm 0.036$ | $\mathbf{0.849 \pm 0.008}$ | $\mathbf{0.787 \pm 0.026}$ | $\mathbf{0.790 \pm 0.011}$ | $\mathbf{0.878 \pm 0.008}$ | $\mathbf{0.821 \pm 0.052}$ | $\mathbf{0.876 \pm 0.011}$ |
| | $\Delta$ | -3.6% | +1.4% | +0.6% | +0.9% | +1.4% | +1.2% | +1.2% |

leverages topological features of attention graphs. All baselines except the unsupervised AttentionScore use the same logistic regression probe architecture and training protocol to ensure fair comparison.

### 3.2. Results

We report results for SinkProbe and the compared baselines in Table 1, where values denote the mean and standard deviation of ROC-AUC over 5-fold cross-validation. Across all evaluated datasets and model families, SinkProbe achieves the best performance in 23 of 28 model-dataset pairs, indicating that sink score-based features provide a robust signal for hallucination detection. The method performs consistently well across diverse tasks and domains (question answering and mathematical reasoning) and across models of varying scale (from 3B up to 12B parameters). Although LookbackLens, MTopDiv, and LapEigval often achieve competitive performance, all three are conceptually related to SinkProbe. In contrast, SinkProbe is a simpler, more general diagnostic: it delivers stronger average performance while identifying sink scores as a simple, direct, and unifying signal underlying several prior attention-based approaches.

## 4. Analysis

To provide deeper insight into SinkProbe, this section analyzes which features drive hallucination detection. Throughout our experiments, we use models of varying sizes from the Mistral and Llama families (Mistral-Nemo and Llama3.2-3B), evaluated on four datasets: NQ-Open and TruthfulQA (question answering), and GSM8K and UMWP (mathematical reasoning).

### 4.1. Sink-Feature Importance in Hallucination Detection

Recall that hallucination detection in SinkProbe is formulated as a classification problem over a feature vector derived from attention sink scores. For each attention head, we retain the top-$k$ largest sink scores and concatenate them across all heads and layers to obtain a fixed-dimensional representation, which is then fed to a logistic regression probe. To determine the importance of each sink score, we examine the probe's learned coefficients. To conduct the analysis, we train the probe with $\ell_1$ regularization to encourage sparsity in the coefficients while maintaining

*Table 2.* Number of non-zero coefficients in the Logistic Regression model trained with $\ell_1$ regularization, i.e., number of sink score features selected for hallucination detection.

| LLM | Dataset | Total Features | Total Important |
|---|---|---|---|
| Llama3.2-3B | GSM8K | 6720 | 38 (1%) |
| | NQ-Open | 6720 | 204 (3%) |
| | TruthfulQA | 6720 | 58 (1%) |
| | UMWP | 6720 | 228 (3%) |
| Mistral-Nemo | GSM8K | 12800 | 82 (1%) |
| | NQ-Open | 12800 | 510 (4%) |
| | TruthfulQA | 12800 | 110 (1%) |
| | UMWP | 12800 | 177 (1%) |

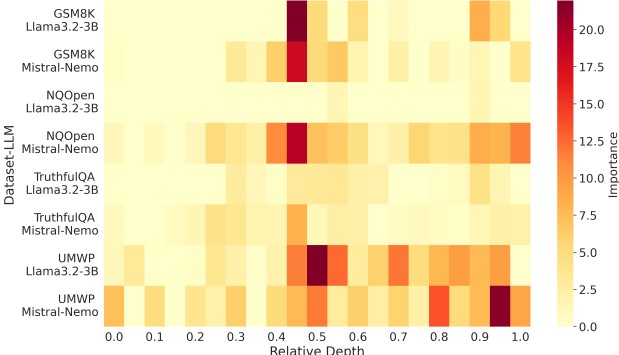

*Figure 4.* Importance scores of attention sinks derived from an $\ell_1$-regularized probe, aggregated across heads and sink indices per layer: $I_l = \sum_h \sum_i |\beta_i^{(l,h)}|$. Scores are plotted against relative layer depth to facilitate cross-model comparison.

comparable performance to the unregularized setting[1]. Let $\beta_i^{(l,h)}$ denote the logistic regression coefficient associated with the $i$-th largest sink score in layer $l$ and head $h$. For each model-dataset pair, we count the number of non-zero coefficients and report the results in Table 2. Notably, the probe retains only a small subset of features—between 1% and 4% of all sink-score features—corresponding to a few dozen to a few hundred coefficients.

Next, we study how sink-score importance is distributed across network depth. For each layer $l$, we define aggregate attention sink importance by summing the absolute values of coefficients associated with that layer,

$$I_l = \sum_h \sum_i \left| \beta_i^{(l,h)} \right|.$$

We visualize the resulting distribution in Figure 4. We find that informative aggregate importance sink scores are dis-

---

[1]To corroborate these findings with an alternative interpretability method, we performed a similar analysis using SHAP (Lundberg & Lee, 2017). While SHAP distributes importance across a broader set of features, the highest-ranked features largely overlap with those identified by the $\ell_1$-regularized model.

tributed across multiple layers, with the middle and final layers contributing most strongly. In light of recent work proposing a mix–compress–refine paradigm of LLM computation (Queipo-de-Llano et al., 2026), this pattern suggests that heightened information mixing in intermediate and final layers may be a key driver of hallucination-related signals. This over-mixing could be coupled with the atypically large value norms, which we examined in Section 2.3.

### 4.2. How Does Top-$k$ Affect Hallucination Detection?

In this study, we compare SinkProbe with other probing approaches that rely on top-$k$ feature selection, namely AttnEigvals and LapEigval. Figure 5 shows how hallucination detection performance varies with increasing $k$. We observe that both LapEigval and SinkProbe achieve their best or near-best performance for small values of $k$ ($k < 10$), whereas raw AttnEigvals improves more gradually with increasing $k$ but does not surpass either method. Notably, SinkProbe exhibits a pronounced performance gain for $k$ between 2 and 5, indicating that hallucination-related information is highly concentrated in a small number of dominant sinks.

This behavior highlights a key advantage of sink-score features over prior attention- and spectral-based representations. While methods such as AttnEigvals and LapEigval rely on more global aggregation of attention structure, sink scores isolate the most extreme attention attractors, yielding a more direct and robust signal. In particular, the leading sink often corresponds to the $\langle \texttt{bos} \rangle$ token (see Figure 3), which already provides a strong hallucination signal, further strengthened by incorporating a small number of additional sinks. Consistent with prior work highlighting the role of attention sinks in information flow (Barbero et al., 2025; Queipo-de-Llano et al., 2026), our results suggest that anomalies in this flow—manifested as disproportionate attention to non-informative tokens—are closely associated with hallucinated outputs. This provides a mechanistic interpretation of why sink-score features capture hallucination-related behavior more effectively than alternative attention-based descriptors.

## 5. Related Work

**Information flow in LLMs** While recent approaches to hallucination detection have focused on exploiting an LLM's internal representations (Chen et al., 2024; Su et al., 2024; Bar-Shalom et al., 2026) or output states (Bar-Shalom et al., 2025; Zhang et al., 2025), there is a growing body of research examining LLM behavior through the lens of information flow. A notable development is the interpretation of attention maps as learned adjacency operators. This perspective creates a formal bridge between Transformers (Vaswani et al., 2017) and Graph Neural Networks (Hamil-

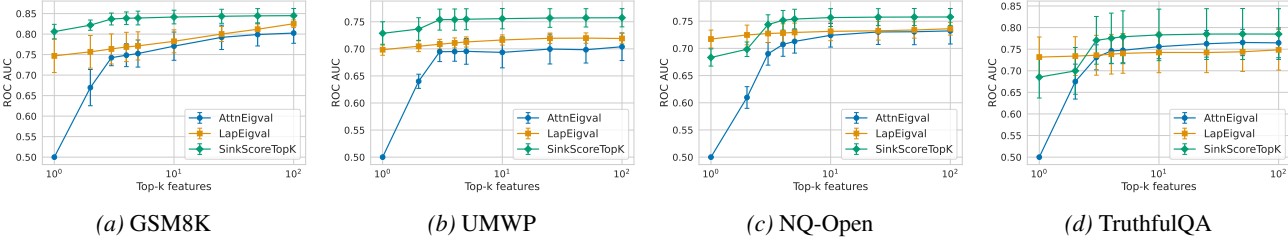

*Figure 5.* Influence of the hyperparameter $k$. Varying the number of retained sink scores per head, we find that SinkProbe attains best or near-best performance even for small values of $k$. This suggests that hallucination-related signals are concentrated in a few dominant sinks and that sink-score features capture this signal more directly and robustly than existing attention- or spectral-based representations. The presented results are for Llama3.2-3B, for other models and datasets see Section C.

ton, 2020), as both architectures propagate information via iterative mixing governed by an underlying graph structure (Arroyo et al., 2025). Barbero et al. (2024; 2025) showed that over-squashing—a bottleneck phenomenon extensively studied in GNNs—can either impair information propagation or prevent excessive over-mixing. Beyond theoretical insights, this graph-theoretic framing opens the door to applying spectral, topological, and message-passing tools to analyze model internals (Binkowski et al., 2025; Bazarova et al., 2025; Frasca et al., 2026). In this work, we use the sink score metric to quantify information flow in LLMs for the purpose of hallucination detection

**Hallucination Detection via Attention Signals** The interpretability afforded by attention-based analysis of Transformer Decoder models has motivated a line of work on detecting hallucinations—defined as fluent outputs unsupported by the input or factual knowledge models (Alansari & Luqman, 2026). Sriramanan et al. (2024) proposed an attention score aggregating diagonal self-attention, observing lower values for hallucinated generations. Chuang et al. (2024) introduced the LookbackLens, contrasting attention to prompt versus response tokens to detect contextual hallucinations. Liu et al. (2025) used attention to partition the input and perform multiple forward passes to assess output consistency. Graph-based approaches have also proven effective: spectral features from Laplacian eigenvalues (Binkowski et al., 2025), topological features (Bazarova et al., 2025; Samaga et al., 2026), and neural message-passing on attention graphs (Frasca et al., 2026) each capture complementary aspects of attention structure. Although these methods were developed from distinct motivations, we show in this work that some of them share a common dependence on attention sink behavior—a unifying perspective that motivates our sink-score-based approach and clarifies the mechanistic basis of attention-based detection.

**Attention Sinks** A striking regularity observed across Transformer architectures is the emergence of *attention sinks*—tokens that attract disproportionate attention mass from subsequent positions regardless of semantic relevance.

First identified in the context of efficient streaming inference (Xiao et al., 2024), attention sinks have since been shown to emerge universally during pretraining (Gu et al., 2025). Rather than a flaw, recent work suggests sinks serve a functional role: Barbero et al. (2025) argue they prevent over-mixing by providing a stable routing target, while Ruscio et al. (2025) offer a geometric characterization of sink formation. Queipo-de-Llano et al. (2026) unify attention sinks with the related phenomenon of compression valleys, showing both arise from massive activations in middle layers and reflect a transition from broad mixing to compressed computation. In this work, we show that a commonly used metric for measuring "sinkness", i.e., the sink score, can provide a rich signal to detect hallucinations.

## 6. Conclusion

We introduce SinkProbe, a simple and effective method for hallucination detection based on attention sink scores. Across a broad range of models and benchmarks, sink-score features provide strong predictive signals and consistently outperform prior attention-based baselines, supporting the view that attention sinks constitute a key internal correlate of hallucination. Beyond raw performance, our analysis shows that the probe relies on a small subset of heads and layers, indicating a relatively localized internal signature that can be inspected and compared across models.

Moreover, we find that this localized signal often correlates with anomalous self-attention value norms, suggesting that hallucinations are closely tied to recent insights into LLM internal dynamics. Overall, we argue that describing information flow in LLMs through the lens of sink scores offers a more fundamental perspective on hallucination detection. In this view, several existing methods can be understood as implicitly leveraging the concept of attention sinks, positioning sink scores as a unifying underlying signal.

An important limitation is that our method requires access to attention weights, which constrains the method to open-weights models. To avoid materializing the full attention matrices, efficient deployment of the method would require

a specialized kernel. Moreover, while sink scores are interpretable, they are a correlational signal and do not, by themselves, establish that sinks are causal drivers of hallucinations. A promising direction is to connect detection with intervention by testing whether manipulating sink formation (e.g., via head ablations or targeted regularization based on attention-output norms) can reduce hallucinations without degrading task performance.

## Impact Statement

Hallucinations in large language models can lead to misinformation, overconfident errors in decision support, and degraded user trust. This work contributes a lightweight detector that can support safer deployment by enabling monitoring, selective abstention, and evaluation of mitigation strategies without relying on external retrieval or repeated sampling. More broadly, our sink-based perspective may help further mechanistic research into how attention routing and information flow relate to hallucinations and other failure modes in LLMs.

## Acknowledgments

We gratefully acknowledge the Wroclaw Centre for Networking and Supercomputing and the Polish high-performance computing infrastructure PLGrid (HPC Center: ACK Cyfronet AGH) for providing computing facilities and support within computational grant no. PLG/2025/018640. This work was co-funded by the National Science Centre, Poland, under CHIST-ERA Open & Re-usable Research Data & Software (grant no. 2022/04/Y/ST6/00183).

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

# A. Methodology and Implementation Details

In our empirical study, we closely follow the methodology of Binkowski et al. (2025), with three key modifications: (1) incorporating the UMWP dataset (Sun et al., 2024) to extend coverage to reasoning tasks, (2) employing cross-validation to improve results reliability, and (3) removing the PCA component to enhance interpretability and facilitate analysis of the trained model (we validated that our conclusions remain consistent regardless of whether PCA is applied).

As a hallucination probe, we selected a logistic regression model. While richer structural features and non-linear probes could further improve performance, the goal of SinkProbe is not to maximize accuracy at all costs, but to test whether attention sink scores alone provide a strong, minimal, and interpretable signal for hallucination detection. Using top-$k$ sink scores with a linear classifier lets us measure how much information is carried by a small number of dominant sinks, without conflating this with gains from more complex architectures or feature transformations. The fact that this simple representation already achieves competitive, and often state-of-the-art, performance suggests that sink scores capture a substantial fraction of the relevant signal.

For logistic regression, we used the scikit-learn implementation (Pedregosa et al., 2011) with `max_iter=1000` and `class_weight=balanced`, leaving all other parameters at their default values. LLM inference was performed using the Transformers library (Wolf et al., 2020) on NVIDIA A40 GPUs and A100 GPUs (48GB and 80GB VRAM, respectively). The implementation of the experiments, including model, prompt, and hyperparameters configurations, is available at github.com/graphml-lab-pwr/sink-probe.

First, we construct hallucinations datasets according to the procedure described in Section B.2. Subsequently, we extract features and use them to train a logistic regression hallucination probe, reporting results for the optimal value of the $k$ hyperparameter. We employ 5-fold cross-validation, ensuring identical fold partitions across all methods for fair comparison.

# B. Datasets Details

## B.1. Upstream Datasets Details

Here, we provide details on the QA and reasoning datasets used to generate hallucination datasets. We selected them based on their prevalence in the literature on hallucination in LLMs. Further details are provided in Table 3.

*Table 3.* Detailed references for the datasets used in the experiments. ([†]Preprocessed following Chen et al. (2024)).

| Dataset | Split / Subset | # Examples | Source |
|---|---|---|---|
| NQ-Open (Kwiatkowski et al., 2019) | Validation | 3,610 | huggingface |
| TriviaQA[†] (Joshi et al., 2017) | Validation | 7,983 | huggingface |
| SQuADv2[†] (Rajpurkar et al., 2018) | Dev (`rc.nocontext`) | 9,960 | official website |
| HaluEvalQA (Li et al., 2023) | QA | 10,000 | official repository |
| TruthfulQA (Lin et al., 2022) | Generation | 817 | huggingface |
| GSM8k (Cobbe et al., 2021) | Test | 1,319 | huggingface |
| UMWP (Sun et al., 2024) | *(entire dataset)* | 5,200 | official repository |

## B.2. Hallucination Datasets Generation

To obtain hallucination labels for all benchmarks,, we ran model inference on each dataset and stored the generated answers together with attention maps. We then used `gpt-4.1` as an LLM-as-judge to compare each answer against the gold reference and assign a hallucination label. For GSM8K, correctness was verified programmatically rather than with the judge. We manually inspected a random subset of judge decisions to ensure label quality. We filtered out responses that exceeded the 2,048-token limit or were deemed unverifiable by the judge (e.g., missing an explicit answer). The remaining examples form the dataset used for training and evaluation.

In order to build our hallucination datasets, we leveraged four common LLMs from three different families, with sizes ranging from 3B up to 12B. We present specific versions of the LLMs in Table 4. For QA datasets we adopted the prompt from Orgad et al. (2025); for GSM8K we used that of `lm-evaluation-harness` (Gao et al., 2024); and for UMWP we used that of Sun et al. (2024). The generation temperature was set to 0.1 and batch size to 1 (He & Lab, 2025).

*Table 4.* LLM details.

| LLM | HuggingFace Repository |
| --- | --- |
| Llama3.2-3B (AI@Meta, 2024) | meta-llama/Llama-3.2-3B-Instruct |
| Llama3.1-8B (AI@Meta, 2024) | meta-llama/Llama-3.1-8B-Instruct |
| Mistral-Nemo (Mistral AI Team & NVIDIA, 2024) | mistralai/Mistral-Nemo-Instruct-2407 |
| Phi3.5 (Abdin et al., 2024) | microsoft/Phi-3.5-mini-instruct |

Table 5 summarizes the resulting sizes and label distributions across datasets and models. From a model-size perspective, invalid counts remain generally low but tend to be slightly higher for smaller models, which more often exceed length limits or omit explicit answers. Hallucination ratios also tend to decrease with scale: larger models are more consistently non-hallucinated across datasets, while smaller models show higher rates.

## C. Optimal $k$ Hyperparameter Values

We run top-$k$ selection among several candidate values $k \in \{1, 2, 3, 4, 5, 10, 25, 50, 100\}$. In Table 6 we present which value of $k$ led to the best results. While we observe that the best mean scores are achieved for the highest value of $k$, similar efficacy is obtained for small values of $k$, as shown in Figure 6.

## D. Semantic Analysis

In Figure 3, we analyzed whether the top tokens, as ranked by sink scores, originated from the prompt or the response. Here, we extend this analysis to examine their semantics. As shown in Figure 7, we identified the three most frequent top-ranked sink tokens in both the prompt (P) and response (R) segments across the two selected datasets and three LLMs. These tokens are grouped by their hallucination label (0 = no hallucination, 1 = hallucination), with counts reflecting their total occurrences. A consistent pattern emerges: regardless of their origin, top sinks are predominantly special tokens (e.g., `<bos>`), whitespace (`<space>`), punctuation, and simple answer-format markers, rather than meaningful content words.

## E. Important Heads Analysis

To identify which attention heads are most predictive of hallucination, we fit an $\ell_1$-regularized logistic regression model on our sink score features. The $\ell_1$ penalty induces sparsity, setting most coefficients $\beta_i$ to exactly zero, thereby performing implicit feature selection. We set $C = 0.75$ based on 5-fold cross-validated ROC-AUC, balancing model sparsity with predictive performance.

**Important Head Selection.** We identify important heads by examining the non-zero coefficients after training logistic regression with $\ell_1$ regularization. For coefficient $\beta_{l,h,k}$ corresponding to layer $l$, head $h$, and top-$k$ position $k$, we compute the odds ratio $\exp(\beta_{l,h,k})$. Features with $|\exp(\beta_{l,h,k}) - 1| > \epsilon$ (where $\epsilon = 10^{-6}$) are retained as important predictors: positive coefficients indicate that higher sink scores increase hallucination probability, while negative coefficients indicate the opposite.

**Layer-wise Importance Aggregation.** To visualize the importance distribution across model depth, we aggregate the absolute coefficient magnitudes by layer:

$$I_l = \sum_{h=1}^{H} \sum_{k=1}^{K} |\beta_{l,h,k}|$$

In our visualizations, we normalize layer indices to relative depth $\delta_l = l/L$ to enable comparison across models with different numbers of layers. The relative depths are discretized into bins of width $0.05$, and importance values are averaged within each bin to produce the heatmap visualization shown in Figure 4.

**Alternative: SHAP-based Selection.** As an alternative to coefficient-based selection, we also considered SHAP values (Lundberg & Lee, 2017) for quantifying feature importance. For each feature, we compute the mean absolute SHAP value across training samples and select features exceeding a specified quantile threshold. Although SHAP provides

*Table 5.* Statistics of the obtained hallucination datasets. #Valid - number of verifiable, valid answers, #Invalid - number of invalid answers (answers which exceeded allowed number of tokens or did not contain answer (judge model marked them unverifiable)), #Non-Hallucinated - number of answers among Valid which were not hallucinated, #Hallucinated - number of answers among Valid which were hallucinated, Hallucination Ratio - ratio of hallucinated answers to the total number of valid answers.

| Dataset | LLM | #Invalid | #Valid Total | Valid Ratio | #Non-Hallucinated | #Hallucinated | Hallucination Ratio |
|---|---|---|---|---|---|---|---|
| TriviaQA | Llama3.1-8B | 368 | 9,592 | 0.96 | 6,423 | 3,169 | 0.33 |
| TriviaQA | Llama3.2-3B | 343 | 9,617 | 0.97 | 5,092 | 4,525 | 0.47 |
| TriviaQA | Mistral-Nemo | 31 | 9,929 | 1.00 | 6,599 | 3,330 | 0.34 |
| TriviaQA | Phi3.5 | 56 | 9,904 | 0.99 | 5,294 | 4,610 | 0.47 |
| NQ-Open | Llama3.1-8B | 731 | 2,879 | 0.80 | 1,056 | 1,823 | 0.63 |
| NQ-Open | Llama3.2-3B | 888 | 2,722 | 0.75 | 1,173 | 1,549 | 0.57 |
| NQ-Open | Mistral-Nemo | 7 | 3,603 | 1.00 | 1,103 | 2,500 | 0.69 |
| NQ-Open | Phi3.5 | 63 | 3,547 | 0.98 | 901 | 2,646 | 0.75 |
| SQuADv2 | Llama3.1-8B | 1,374 | 4,554 | 0.77 | 1,081 | 3,473 | 0.76 |
| SQuADv2 | Llama3.2-3B | 916 | 5,012 | 0.85 | 898 | 4,114 | 0.82 |
| SQuADv2 | Mistral-Nemo | 20 | 5,908 | 1.00 | 1,327 | 4,581 | 0.78 |
| SQuADv2 | Phi3.5 | 173 | 5,755 | 0.97 | 1,388 | 4,367 | 0.76 |
| TruthfulQA | Llama3.1-8B | 94 | 723 | 0.88 | 234 | 489 | 0.68 |
| TruthfulQA | Llama3.2-3B | 54 | 763 | 0.93 | 200 | 563 | 0.74 |
| TruthfulQA | Mistral-Nemo | 4 | 813 | 1.00 | 207 | 606 | 0.75 |
| TruthfulQA | Phi3.5 | 12 | 805 | 0.99 | 243 | 562 | 0.70 |
| HaluEvalQA | Llama3.1-8B | 2,468 | 7,532 | 0.75 | 2,551 | 4,981 | 0.66 |
| HaluEvalQA | Llama3.2-3B | 2,269 | 7,731 | 0.77 | 2,021 | 5,710 | 0.74 |
| HaluEvalQA | Mistral-Nemo | 34 | 9,966 | 1.00 | 3,492 | 6,474 | 0.65 |
| HaluEvalQA | Phi3.5 | 209 | 9,791 | 0.98 | 2,788 | 7,003 | 0.72 |
| UMWP | Llama3.2-3B | 5 | 5,195 | 1.00 | 4,021 | 1,174 | 0.23 |
| UMWP | Llama3.1-8B | 55 | 5,145 | 0.99 | 4,348 | 797 | 0.15 |
| UMWP | Mistral-Nemo | 6 | 5,194 | 1.00 | 4,399 | 795 | 0.15 |
| UMWP | Phi3.5 | 182 | 5,018 | 0.96 | 4,111 | 907 | 0.18 |
| GSM8K | Llama3.2-3B | 17 | 1,302 | 0.99 | 973 | 329 | 0.25 |
| GSM8K | Llama3.1-8B | 22 | 1,297 | 0.98 | 1,104 | 193 | 0.15 |
| GSM8K | Mistral-Nemo | 108 | 1,211 | 0.92 | 1,058 | 153 | 0.13 |
| GSM8K | Phi3.5 | 162 | 1,157 | 0.88 | 1,019 | 138 | 0.12 |

model-agnostic importance scores that account for feature interactions, we found that $\ell_1$-regularized coefficients yield sparser selections, and that the highest-ranked SHAP features overlap with those identified by the logistic regression model.

*Table 6.* Values of top-$k$ providing best quality

| LLM | Dataset Feature | GSM8K | HaluEvalQA | NQ-Open | SQuADv2 | TriviaQA | TruthfulQA | UMWP |
|---|---|---|---|---|---|---|---|---|
| Llama3.2-3B | AttnEigval | 100 | 100 | 100 | 100 | 100 | 50 | 100 |
| | LapEigval | 100 | 100 | 100 | 100 | 100 | 100 | 50 |
| | SinkProbe | 100 | 50 | 100 | 25 | 100 | 50 | 100 |
| Phi3.5 | AttnEigval | 100 | 50 | 100 | 100 | 100 | 100 | 100 |
| | LapEigval | 10 | 50 | 100 | 100 | 100 | 100 | 100 |
| | SinkProbe | 50 | 50 | 50 | 100 | 50 | 2 | 50 |
| Llama3.1-8B | AttnEigval | 100 | 100 | 100 | 100 | 100 | 100 | 100 |
| | LapEigval | 100 | 100 | 100 | 100 | 100 | 10 | 100 |
| | SinkProbe | 3 | 100 | 100 | 25 | 100 | 100 | 100 |
| Mistral-Nemo | AttnEigval | 100 | 100 | 100 | 100 | 50 | 100 | 100 |
| | LapEigval | 100 | 25 | 50 | 100 | 25 | 100 | 100 |
| | SinkProbe | 2 | 50 | 100 | 100 | 25 | 100 | 100 |

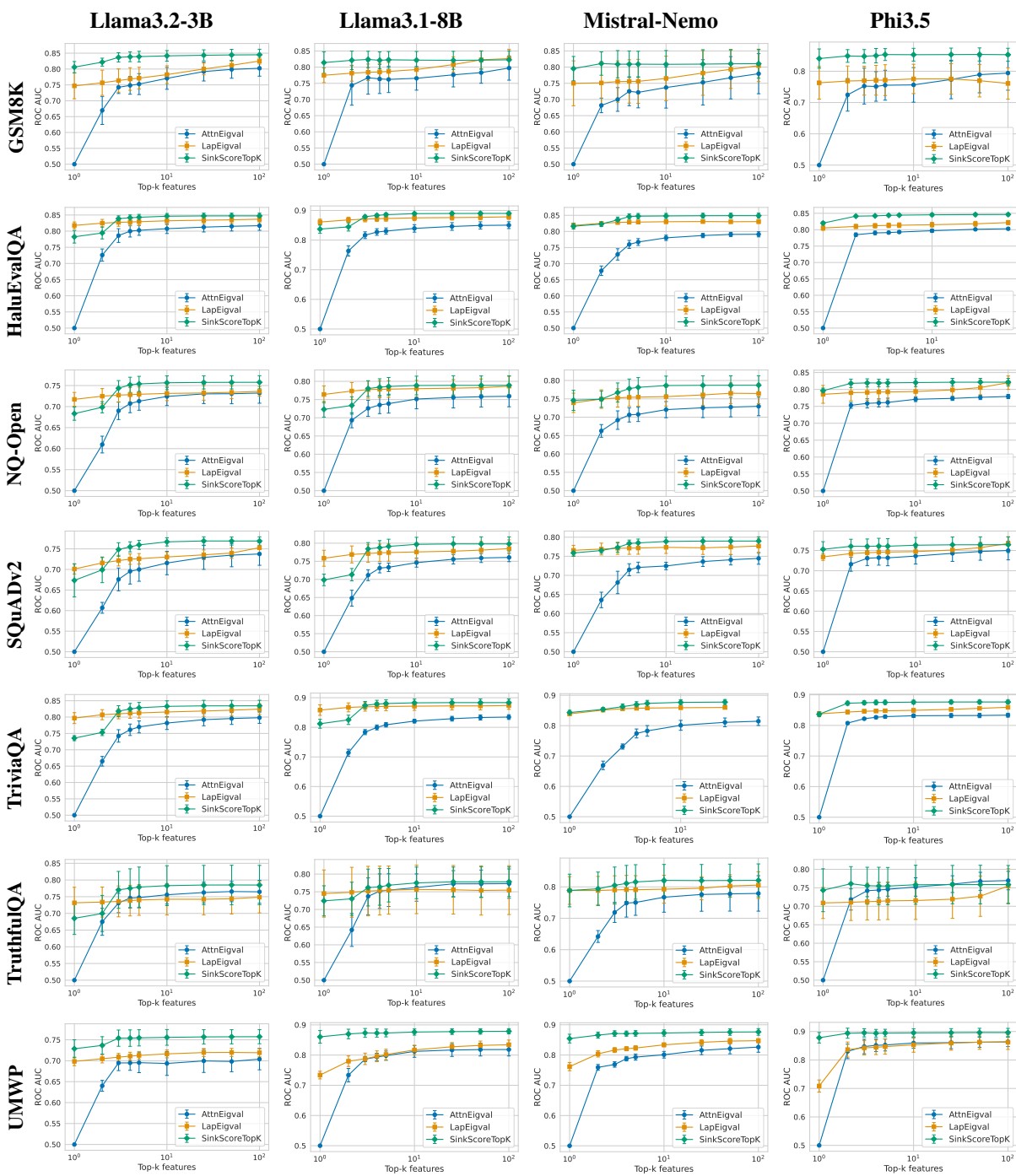

*Figure 6.* Hallucination detection performance (ROC-AUC) as a function of top-$k$ across all models and datasets. Each subplot shows the ROC-AUC performance on 5-fold cross-validation for different values of $k \in \{1, 2, 3, 4, 5, 10, 25, 50, 100\}$ and models. Columns represent different LLMs, rows represent different datasets.

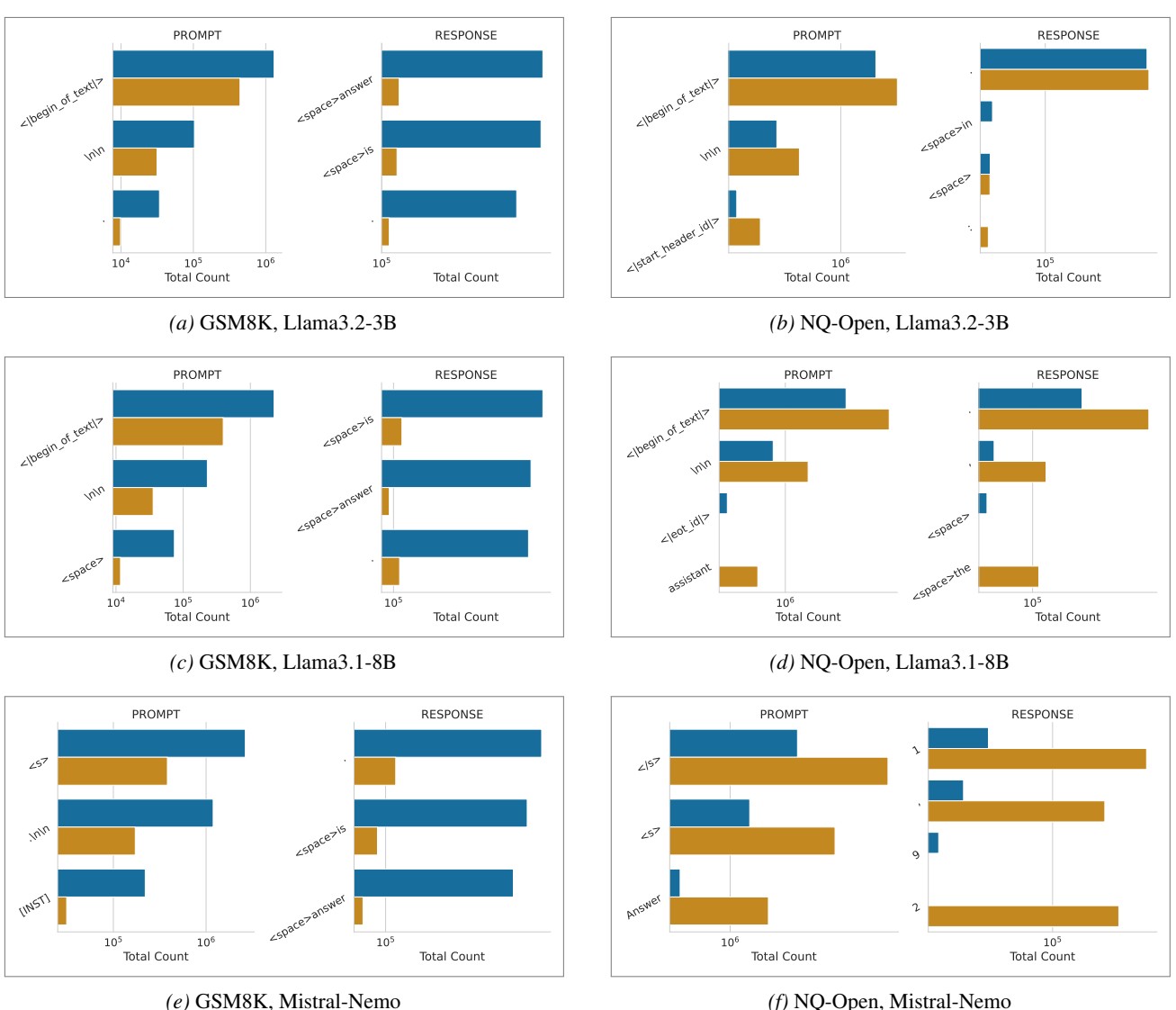

*Figure 7.* Token-semantic characterization. Each subplot shows the 3 most frequent tokens in the prompt and response segments of the top-ranked attention sinks, grouped by hallucination label: non-hallucinated and hallucinated. Counts reflect the total number of tokens' occurrences.

