# OpenReview forum: "Attention Sinks as Internal Signals for Hallucination Detection in Large Language Models"
_ICML.cc/2026/Conference — ICML 2026 regular_

### Official Review · Reviewer_4DDX · 2026-03-05

**Soundness:** 3
**Presentation:** 2
**Significance:** 3
**Originality:** 3
**Overall Recommendation:** 4
**Confidence:** 4

**Summary:**

This study investigates the hallucination detection question in LLMs through the analysis of internal attention dynamics. Specifically, it presents SinkProbe, a straightforward feature-extraction and probing method built upon attention sink scores.
The work establishes that tokens accumulating excessive attention mass during generation—defined as attention sinks—act as a strong correlational signal for hallucinations, especially when paired with high-norm value vectors.
The proposed method computes attention sink scores across all layers and attention heads, extracts top-k values, and uses them as input features for a logistic regression classifier.
Extensive experiments are conducted on four open-weight LLMs over seven datasets, with results showing improved ROC-AUC performance compared to recent attention-based baseline methods.

**Compliance With Llm Reviewing Policy:**

Affirmed.

**Key Questions For Authors:**

See the Weaknesses.

**Limitations:**

Yes

**Strengths And Weaknesses:**

Strengths:

1. The paper conducts a thorough evaluation across a solid variety of recent open-weight models and diverse datasets encompassing both knowledge-intensive QA and mathematical reasoning.

2. The observation linking "computationally active sinks" to hallucination behavior is a neat finding. It adds a valuable nuance to the existing literature on attention sinks, which have traditionally been viewed just as semantic "dumping grounds".

3. The proposed SinkProbe method consistently outperforms other attention-based baselines across the majority of the tested model-dataset configurations.

Weaknesses:

1. In Section 2.4, the authors claim to "unify" existing detection methods by mathematically relating them to sink scores. However, demonstrating that the diagonal of a Laplacian matrix algebraically incorporates the column sum minus self-attention is a basic property of graph Laplacians. This is an algebraic equivalence rather than a profound mechanistic unification, and it does not definitively prove that the success of spectral or topological methods is solely driven by sink tokens.

2. The authors frame their method as a "lightweight detector" in the Impact Statement. However, computing sink scores requires full access to the T×T attention matrices across all layers and heads at inference time. In real-world deployment, materializing and aggregating these attention matrices introduces massive memory bandwidth and computational overheads. This significantly hinders its practical utility compared to standard output-probability or entropy-based uncertainty metrics, which are surprisingly absent from the baseline comparisons.

3. The methodology relies heavily on an LLM-as-a-judge to generate the binary hallucination labels for training and testing the probe. Because LLM judges are known to exhibit specific biases, length preferences, and their own hallucination patterns, it is highly likely that the linear probe is simply learning to predict GPT-4.1's latent preferences or certainty rather than objective factual truth.

---

> ### Author Rebuttal · Authors · 2026-03-31
>
> We thank the reviewer for highlighting the paper’s strengths and for constructive comments.
>
> **On the diagonal of a Laplacian matrix**
> We agree that our paper establishes the algebraic equivalence between Laplacian eigenvalues and sink scores. However, we believe this observation is both important and novel, as it has not been identified in prior literature. First, it reveals a connection between related methods and suggests that they may rely on a common underlying signal for hallucination detection. Moreover, our experiments show a consistent improvement of SinkProbe over LapEigval, suggesting that sink scores may capture a more direct and informative signal associated with model hallucinations. Our work doesn’t fully prove that hallucinations are solely driven by sink score anomalies, but empirically shows that they are manifested whenever hallucination occurs.
>
> **On computational overhead and practicality.**
> We agree that the current method is less deployment-friendly than output-probability or entropy-based metrics, because it requires access to attention weights. In our present implementation, naively materializing $T \times T$ attention matrices would indeed incur substantial overhead. However, the method fundamentally requires only sink-score accumulations, which could be integrated into custom attention kernels and computed online during decoding without storing full attention matrices. We have not developed such an optimized implementation in this work, and regard it as future engineering work needed for productization. Accordingly, our use of “lightweight” is meant relative to prior detectors relying on multiple generations or heavy detectors, for which *SinkProbe* remains more efficient.
>
>
> **On evaluation protocol**
> We agree that LLM-based judges can exhibit biases (e.g., length preferences or their own hallucination patterns), and that this is a general limitation of recent hallucination detection benchmarks.
> Importantly, we also evaluate on settings with objective ground-truth signals. In particular, in Section 4.1 we manually verified a subset of examples, consistent with commonly adopted protocols in the literature. Additionally, for datasets such as GSM8K, we rely on automatic correctness evaluation, ensuring objective and reproducible measurements.
> More broadly, we note that the use of LLM-as-a-judge supervision is common across prior work [1,2,3,4] in hallucination detection.  Our method therefore operates under the same evaluation protocol, enabling fair comparison. Also, we are working towards including the suggested baselines into the results, yet we want to highlight that these methods operate on the different features (final logits/probabilities instead of attention maps).
>
> **References**
>
> [1] Gaurang Sriramanan, Siddhant Bharti, Vinu Sankar Sadasivan, Shoumik Saha, Priyatham Kattakinda, and Soheil Feizi. 2024. *LLM-Check: Investigating Detection of Hallucinations in Large Language Models*. NeurIPS 2024 poster.
>
> [2] Jakub Binkowski, Denis Janiak, Albert Sawczyn, Bogdan Gabrys, and Tomasz Jan Kajdanowicz. 2025. *Hallucination Detection in LLMs Using Spectral Features of Attention Maps*. In *Proceedings of the 2025 Conference on Empirical Methods in Natural Language Processing*, pages 24354–24385, Suzhou, China. Association for Computational Linguistics.
>
> [3] Hadas Orgad, Michael Toker, Zorik Gekhman, Roi Reichart, Idan Szpektor, Hadas Kotek, and Yonatan Belinkov. 2024. *LLMs Know More Than They Show: On the Intrinsic Representation of LLM Hallucinations*. arXiv:2410.02707.
>
> [4] Fabrizio Frasca, Guy Bar-Shalom, Yftah Ziser, and Haggai Maron. 2025. *Neural Message-Passing on Attention Graphs for Hallucination Detection*. arXiv:2509.24770.

---

### Official Review · Reviewer_vrBG · 2026-03-10

**Soundness:** 4
**Presentation:** 4
**Significance:** 4
**Originality:** 3
**Overall Recommendation:** 4
**Confidence:** 4

**Summary:**

This paper proposes a simple hallucination detector for LLMs based on “attention sinks” (tokens that attract unusually large attention during generation). It summarizes each response by the top-K sink scores from the model’s attention maps and feeds them into a lightweight classifier, showing strong detection performance across several models and benchmarks, plus some analysis of which sinks matter most.

**Compliance With Llm Reviewing Policy:**

Affirmed.

**Key Questions For Authors:**

See my weakness section

**Limitations:**

Yes

**Strengths And Weaknesses:**

Strengths:

* This paper is well written and easy to follow.

* The experimental setup is very comprehensive, including multiple LLMs, baselines, and datasets.

* The authors attempt to show how their method connects to prior work (e.g., LookBackLens and LLMCheck), which helps readers better understand the field rather than simply claiming “we are better than the competition.”

* In addition to the main results, several analyses shed light on the importance of attention sinks as indicators of hallucinations.

Weaknesses:

* I’m not sure this is a weakness, but it feels like the authors leave a lot of performance on the table by taking the top-K attention sinks and concatenating them before feeding them into a logistic regression model. While I have seen this design choice in other work in the field [1], there are also works showing that preserving structural information and going beyond linear probes can significantly improve results [2]. (Please cite both papers, as they are missing from your paper.)

* Related to the previous point, another concern is that the improvements appear incremental in many cases (and I appreciate the authors including confidence intervals). Adding an average (“avg”) column to Table 1 would help readers assess this more easily.

* Demonstrating something stronger than correlation would be very interesting here (and I appreciate the authors mentioning this in the future work section).

*References:*


[1] https://arxiv.org/pdf/2503.14043

[2] https://arxiv.org/pdf/2510.00296

---

> ### Author Rebuttal · Authors · 2026-03-31
>
> We thank the reviewer for the thoughtful and constructive assessment. We appreciate the recognition of the paper’s clarity, the breadth of the experimental evaluation, and our effort to connect the method to prior work. We also value the reviewer’s suggestions on stronger modeling choices, clearer aggregate reporting, and the importance of moving beyond correlational evidence.
>
> **On the linear probe**
> We agree that richer structural features and non-linear probes could further improve performance. However, this design choice is intentional: the goal of $SinkProbe$ is not to maximize accuracy at all costs, but to test whether attention sink scores alone provide a strong, minimal, and interpretable signal for hallucination detection.
> Using top-$k$ sink scores with a linear classifier lets us measure how much information is carried by a small number of dominant sinks, without conflating this with gains from more complex architectures or feature transformations. The fact that this simple representation already achieves competitive, and often state-of-the-art, performance suggests that sink scores capture a substantial fraction of the relevant signal.
> We agree that preserving richer structure, such as full attention patterns or interactions between sinks, is a promising direction for future work. We will add this discussion and cite the suggested papers in the revised version.
>
> **On the gains**
> We would like to emphasize that across the evaluated settings, our method outperforms baselines in the majority of cases, i.e., we outperformed baselines in 23 out of 28 cases while remaining competitive elsewhere. However, in this research we show that many of the compared approaches operate on closely related attention-derived signals, which, we suppose, inherently limits the achievable performance gap. In this context, consistently matching or surpassing these methods with a significantly simpler representation is a meaningful result. These observations are also aligned with our analysis, which shows that much of the predictive signal is concentrated in a small number of dominant sinks. Also, we will revise the table and add the suggested "average column" for better results presentation.
>
> **On the results beyond correlation**
> We agree that going beyond correlation would be an interesting and valuable direction. At the same time, we view this as complementary to the main contribution of the paper, which is to identify sink scores as a strong internal signature of hallucination. In general, an informative predictive signal does not necessarily imply a simple or effective intervention, especially when the relevant behavior is likely distributed across tokens, heads, and layers rather than tied to a single easily manipulable factor. We therefore see stronger causal validation as an important direction for future work, and will clarify this more explicitly in the revised version. We are also exploring more targeted intervention strategies, and will include additional results if they become available during the discussion period.

---

> > ### Author Rebuttal · Reviewer_vrBG · 2026-04-02
> >
> > The authors provided decent answers, so I will keep my (relatively) high score.
> > Good luck!

---

### Official Review · Reviewer_9Jg5 · 2026-03-11

**Soundness:** 3
**Presentation:** 2
**Significance:** 2
**Originality:** 3
**Overall Recommendation:** 4
**Confidence:** 2

**Summary:**

The paper proposes SinkProbe, a hallucination detection method that uses attention sink scores as features for a logistic regression classifier. For each layer and head, the method computes token-wise sink scores from attention matrices, takes the top-k order statistics, and concatenates them into a feature vector for binary classification of generations as hallucinated vs non-hallucinated. The authors further argue that only a small subset of "computationally active" sinks (those associated with large value norms) drive hallucinations, and they provide a unifying interpretation of several existing attention-based detectors as transformations of sink behavior. Experiments across 4 open-weight LLMs and 7 hallucination benchmarks show that SinkProbe usually yields slightly higher ROC-AUC than prior attention-based baselines.

**Compliance With Llm Reviewing Policy:**

Affirmed.

**Final Justification:**

I will maintain my score

**Key Questions For Authors:**

please refer to weaknesses

**Limitations:**

please refer to weaknesses

**Strengths And Weaknesses:**

Strengths:

1.Clear, simple method built on a principled internal signal.

2.Systematic empirical comparison to several attention-based baselines.

3.The writing is clear and easy to follow.

Weaknesses:

1.The analysis in Figure 3 on the location of top-k sinks is informative, showing that the top-ranked sink is almost always in the prompt (likely <bos>) and lower-ranked sinks more often in the response. However, the paper stops at quantifying positions and does not examine, for example, what types of tokens (punctuation, numbers, named entities) correspond to those critical sinks, or give qualitative examples of attention patterns in hallucinated vs grounded answers. This would be particularly useful for judging whether the signal is semantically meaningful or largely tied to special-token idiosyncrasies.

2.The empirical gains are promising but not fully convincing yet. SinkProbe is often competitive, but many improvements over the strongest baselines are fairly small, and in several settings it is not the best method.

3.The paper covers several relevant attention-based detectors but omits other closely related work, especially on internal-state-based hallucination detection and attention-guided methods. The following should be added and discussed:

[1] Liu et al., "Attention-Guided Self-Reflection for Zero-Shot Hallucination Detection in Large Language Models", 2025.

[2] Zhang L. et al., "Detecting Hallucination in Large Language Models Through Deep Internal Representation Analysis (MHAD)", 2025.

[3] Unsupervised Real-Time Hallucination Detection based on the Internal States of Large Language Models (MIND)", 2024

---

> ### Author Rebuttal · Authors · 2026-03-31
>
> We thank the reviewer for their thoughtful evaluation and constructive feedback. We appreciate the recognition of SinkProbe's principled design, systematic comparison, and clear writing. Below we address raised concerns.
>
> **On the empirical gains.**
> We agree that the margins are sometimes modest and that SinkProbe is not the top method in every setting. However, SinkProbe outperforms the baselines in 23 of 28 evaluated settings and remains competitive in the others, despite being a simpler and more general diagnostic. Since many compared methods rely on closely related attention-based signals, large performance gaps are not necessarily expected - achieving similar or better performance with a minimal representation is itself a meaningful result.
> More importantly, our main contribution is not only stronger average performance, but the identification of sink scores as a simple, direct, and unifying signal underlying several prior attention-based approaches. We hope this perspective will guide future detectors toward more interpretable and lightweight internal signals, and potentially inform training-time interventions for reducing hallucinations.
>
> **On the semantic analysis.**
> We conducted an additional study examining token types associated with top-$k$ sink scores. For two LLMs and datasets, we identified the three most frequent tokens among top-$k$ sinks in both the prompt (P) and response (R) segments, grouped by label (0 = no hallucination, 1 = hallucination). Counts reflect the number of tokens' occurences. A consistent pattern emerges: top sinks are predominantly special tokens (e.g., $\langle bos \rangle$), whitespace (`<space>`), punctuation, and simple answer-format markers - not content words. This confirms that the predictive signal reflects atypical attention patterns governing information flow rather than lexical content. We will include additional qualitative examples and token-type statistics in the final version.
>
> **Llama-3.1-8B**
>
> | Label | Seg. | Token | Count | | Label | Seg. | Token | Count |
> |------:|:-----|:------|------:|---|------:|:-----|:------|------:|
> | | | *GSM8K* | | | | | *NQOpen* | |
> | 0 | P | `<\|begin_of_text\|>` | 2.25M | | 0 | P | `<\|begin_of_text\|>` | 2.16M |
> | 0 | P | `<space>` | 229K | | 0 | P | `<space>` | 857K |
> | 0 | P | `<space>` | 74K | | 0 | P | `<\|eot_id\|>` | 480K |
> | 0 | R | `is` | 638K | | 0 | R | `.` | 184K |
> | 0 | R | `answer` | 551K | | 0 | R | `,` | 62K |
> | 0 | R | `.` | 534K | | 0 | R | `<space>` | 57K |
> | 1 | P | `<\|begin_of_text\|>` | 394K | | 1 | P | `<\|begin_of_text\|>` | 3.72M |
> | 1 | P | `<space>` | 36K | | 1 | P | `<space>` | 1.33M |
> | 1 | P | `<space>` | 12K | | 1 | P | `assistant` | 706K |
> | 1 | R | `is` | 111K | | 1 | R | `.` | 420K |
> | 1 | R | `.` | 108K | | 1 | R | `,` | 118K |
> | 1 | R | `answer` | 95K | | | | | |
>
> **Llama-3.2-3B**
>
> | Label | Seg. | Token | Count | | Label | Seg. | Token | Count |
> |------:|:-----|:------|------:|---|------:|:-----|:------|------:|
> | | | *GSM8K* | | | | | *NQOpen* | |
> | 0 | P | `<\|begin_of_text\|>` | 1.30M | | 0 | P | `<\|begin_of_text\|>` | 1.57M |
> | 0 | P | `<space>` | 103K | | 0 | P | `<space>` | 437K |
> | 0 | P | `.` | 34K | | 0 | P | `<\|start_header_id\|>` | 260K |
> | 0 | R | `answer` | 344K | | 0 | R | `.` | 220K |
> | 0 | R | `is` | 339K | | 0 | R | `in` | 66K |
> | 0 | R | `.` | 281K | | 0 | R | `<space>` | 65K |
> | 1 | P | `<\|begin_of_text\|>` | 441K | | 1 | P | `<\|begin_of_text\|>` | 2.08M |
> | 1 | P | `<space>` | 31K | | 1 | P | `<space>` | 585K |
> | 1 | P | `.` | 10K | | 1 | P | `<\|start_header_id\|>` | 353K |
> | 1 | R | `answer` | 114K | | 1 | R | `.` | 224K |
> | 1 | R | `is` | 112K | | 1 | R | `<space>` | 65K |
> | 1 | R | `.` | 106K | | 1 | R | `:` | 64K |
>
> **Relation to related work.**
> Liu et al. (2025) use attention to partition input and perform multiple forward passes to assess output consistency. Our method instead extracts a diagnostic signal in a single forward pass, showing that sink scores alone capture much of the hallucination-related signal without re-generation overhead.
>
> Zhang et al. (MHAD, 2025) detect hallucinations using statistical features of token probabilities. Our work is complementary - we operate on internal attention dynamics rather than output probabilities, capturing fundamentally different aspects of model behavior: sink scores reflect internal information flow, while token likelihoods characterize surface-level output distributions.
>
> Chen et al. (MIND, 2024) train a classifier over high-dimensional internal states including activations and attention patterns. We show that a minimal, interpretable subset - simple sink score statistics - suffices to achieve competitive or state-of-the-art performance, suggesting the predictive signal is concentrated in a few dominant attention sinks.
>
> We will add discussion of all three works to the related work section in the final version.

---

> > ### Author Rebuttal · Reviewer_9Jg5 · 2026-04-04
> >
> > Thanks for the reply. Most of my concerns are addressed. I will keep my score

---

### Official Review · Reviewer_6dpV · 2026-03-13

**Soundness:** 3
**Presentation:** 4
**Significance:** 3
**Originality:** 3
**Overall Recommendation:** 4
**Confidence:** 4

**Summary:**

This paper demonstrates that large language model hallucinations are deeply connected to *attention sinks*, which are tokens that accumulate disproportionate attention during text generation. To leverage this mechanism, the authors introduce SinkProbe, a lightweight and resource-efficient detector that achieves state-of-the-art hallucination detection using only the model's internal attention maps. The study reveals that the most critical predictive signals come from *computationally active* sinks that possess unusually large value vector norms. Ultimately, the research unifies the field of attention-based hallucination detection by mathematically proving that prior existing methods implicitly depend on this exact same attention sink behavior. Empirically, experiments across multiple model families and benchmarks confirm that SinkProbe consistently outperforms existing baselines, proving that sink-score features provide a highly robust and versatile signal for identifying hallucinations.

**Compliance With Llm Reviewing Policy:**

Affirmed.

**Final Justification:**

While the limitation regarding black-box models remains, the authors have appropriately framed the work as a mechanistic diagnostic tool for open-weights models. Given the thoroughness of the empirical results and the unifying theoretical contribution, I raised my score from 3 to 4.

**Key Questions For Authors:**

1. Have you conducted intervention experiments, such as targeted attention masking, value vector normalization, or specific head ablations on these sink tokens, during decoding to see if it actively mitigates or prevents the hallucinated output? Analyses on these experiments would make the conclusion more convincible.

2. Is there a consistent positional or semantic pattern to these highly informative sinks across different prompts, and could identifying these patterns eventually help approximate this diagnostic for black-box models?

**Limitations:**

Yes

**Strengths And Weaknesses:**

### **Strengths**

The paper introduces a compelling framework for hallucination detection through the lens of attention sinks. It successfully unifies several disparate attention-based methods under this single conceptual umbrella. The authors clearly demonstrate how previous structural diagnostics inherently rely on attention sink behavior.

The experimental design is thorough, evaluating the approach across seven diverse benchmarks spanning question answering and mathematical reasoning. The study robustly tests the method on four distinct open-weight base models ranging from 3B to 12B parameters. Furthermore, by reporting the mean and standard deviation of ROC-AUC scores over 5-fold cross-validation, the authors provide strong statistical reliability for their performance claims.

### **Weaknesses**

The paper primarily observes a correlational phenomenon without fully explaining the theoretical mechanism behind why attention sinks cause hallucinations. The submission would be substantially stronger with deeper theoretical analysis or causal experiments, such as targeted head ablations, to prove that sinks are direct drivers of fabricated outputs rather than just symptoms.

The proposed method is practically restricted to open-source, white-box models because it strictly requires access to the model's internal attention weights. This structural limitation prevents its application in scenarios where researchers or practitioners only have black-box access to the model.

---

> ### Author Rebuttal · Authors · 2026-03-31
>
> We thank the reviewer for the positive assessment of the paper, particularly for recognizing the clarity of the framework, the strength of the empirical evaluation, and the unifying perspective on attention-based hallucination detection.
>
> **Causal experiments**
> We agree that causal validation through interventions would strengthen the mechanistic interpretation. To probe this, we experimented with several strategies that suppress tokens with high sink scores. While these interventions did not produce significant improvements, we do not view this as contradicting the main contribution. First, predictive usefulness and controllability are distinct: a signal can be highly informative about hallucination without immediately yielding an effective decoding-time control method. Second, sink scores are most reliably identified only after the full generation is observed, which makes online intervention inherently approximate and temporally misaligned. Third, hallucinations are likely driven by distributed computations across tokens, heads, and layers, so token suppression is a relatively coarse intervention that may not isolate the underlying causal mechanism. We therefore interpret the current results as evidence that sink scores are a meaningful mechanistic signature of hallucination, while stronger causal interventions remain an important direction for future work. Nonetheless, we are currently exploring more targeted intervention strategies (e.g., head- and layer-specific modulation and temporally aligned suppression), and will include additional results if they become available during the rebuttal period.
>
>
> **On the theoretical mechanism behind attention sinks.**
> We agree that understanding why attention sinks are linked to hallucinations is an important aspect. In the current version, we focus on identifying a robust internal signal; however, we do provide a mechanistic interpretation based on the interaction between attention weights and value vectors. In particular, while sink scores capture the concentration of attention, the actual influence on the model’s computation is governed by the product $A V$. Our analysis shows that only a subset of sinks—those associated with large value norms—are computationally active and can dominate the residual stream through repeated injection of high-magnitude signals. This provides a concrete explanation of how sink behavior can lead to prior-dominated or hallucinated outputs. We will further clarify and strengthen this explanation in the revised version.
>
> **On applicability to black-box models.**
> We agree that the current method is restricted to white-box models, as it relies on access to attention weights. However, this does not diminish its value as an analysis tool: beyond detection performance, our method identifies stable and interpretable sink patterns—both across token positions and across a small subset of heads and layers—that appear closely linked to hallucination behavior in transformer decoders. We therefore see the main contribution not only as a practical detector for open models, but also as evidence about internal mechanisms that could motivate future black-box approximations. We will clarify this limitation and frame the contribution more explicitly in the paper.
>
> **On positional and semantic patterns of sinks.**
> We appreciate this suggestion. Beyond the positional analysis already presented, we further examined the highest-ranked sink tokens and found a consistent pattern across datasets and models: they are predominantly special tokens (e.g., $\langle bos \rangle$), whitespace, punctuation, and simple answer-format markers. This observation suggests that the predictive signal is not driven mainly by specific content words, but instead reflects atypical attention patterns that influence information flow in the LLM. We will include additional qualitative examples and token-type statistics in the final version to further illustrate the largely structural character of these sinks. Due to space limitation, we put the table presenting the results in response to `Reviewer 9Jg5`.
>
> **Summary.**
> We thank the reviewer again for the constructive feedback. We will strengthen the mechanistic explanation, expand the analysis of sink token types, and clarify the scope and limitations of the method in the revised version.

---

> > ### Author Rebuttal · Reviewer_6dpV · 2026-04-04
> >
> > Thank the authors for their detailed rebuttal and for conducting additional intervention experiments. I found the explanation regarding the AV interaction, specifically how large value norms render certain sinks "computationally active"—to be a convincing step toward bridging the gap between correlation and causation.
> >
> > Furthermore, the clarification regarding the semantic patterns of sinks provides helpful intuition on why these signals are robust across different prompts. While the limitation regarding black-box models remains, the authors have appropriately framed the work as a mechanistic diagnostic tool for open-weights models. Given the thoroughness of the empirical results and the unifying theoretical contribution, I will adjust my score to reflect a positive recommendation.

---

### Decision · Program_Chairs · 2026-04-30

**Decision:**

Accept (regular)

**Comment:**

This paper proposes SinkProbe, a hallucination detector based on attention sinks in LLMs. Reviewers agreed that the paper is clearly written, technically solid, and supported by a broad empirical evaluation across multiple models, datasets, and baselines. They also found the attention-sink perspective interesting and useful, especially in providing a unifying view of prior attention-based detection methods.

The main remaining concerns are that the paper does not yet establish a strong causal mechanism behind the observed correlation, that the method is limited to white-box settings with access to internal attention, and that some empirical gains over prior methods are modest. Reviewers also noted that the broader unification claim should be interpreted with some caution.

After the rebuttal, however, the concerns were largely addressed or clarified. In particular, the authors better framed the contribution as a mechanistic diagnostic tool for open-weight models rather than a universally deployable detector, and added useful analysis supporting the role of computationally active sinks. Given the overall strength of the empirical study and the clarity of the contribution, I find the paper to make a meaningful contribution to LLM hallucination analysis and interpretability.

I therefore recommend acceptance. For the final version, the authors should further clarify the scope and limitations of the method, temper causal claims where appropriate, and incorporate the additional related-work and analysis promised in the rebuttal.